

# Effects of stumping on fine root architecture, growth, and physiology of *Hippophae rhamnoides*

Haoyue Wang[1,*], Wei Qi[2,*], Yuefeng Guo[1] and Yajie Xu[1]

[1] College of Desert Control Science and Engineering, Inner Mongolia Agricultural University, Hohhot, Inner Mongolia, China

[2] Inner Mongolia Autonomous Region Water Conservancy Development Center, Hohhot, Inner Mongolia, China

[*] These authors contributed equally to this work.

Corresponding author
Yuefeng Guo, bfj@emails.imau.edu.cn

## ABSTRACT

**Background**. Fine roots are vital to a plant's ability to absorb water and nutrients. Stumping is a practice that may encourage fine root growth and the rapid recovery of decaying *Hippophae rhamnoides* plants. However, the effect of stumping on the fine roots and physiological indices is still unknown. The differential indices between stumped forests and non-stumped forests must also be defined.

**Methods**. We recorded the changes in the fine roots of structure *H. rhamnoides* one year after stumping. Using single factor analysis of variance and general linear models we comprehensively analyzed the number of root tips and the plant's growth and physiological indices in response to stumping. Partial least squares discriminant analysis (PLS-DA) was used to compare fine root growth and physiological indices with and without stumping in order to identify the differential indices.

**Results**. The proportion of root tips in the vertical layers at 30–40 cm and 40–50 cm and in the horizontal layers at 60–80 cm and 80–100 cm, increased after stumping by 1.85%, 2.60%, 1.96% and 4.32%, respectively. In the 0–50 cm soil layer, the fine root dry weight rose by 27.6% compared with the control, which was not significant. However, other indices were significantly different from the control. The proportions in the growth indices in the 30–40 cm and 40–50 cm layers increased after stumping. Stumping had a significant, negative effect on proline and malondialdehyde content, which dropped by 40.95% and 55.32%, respectively, indicating that the harms caused by these two chemicals was alleviated. Stumping had a significant positive effect on root activity and soluble sugar contents, which increased by 68.58% and 36.87%, respectively, and improved the growing ability of fine roots. PLS-DA revealed that malondialdehyde, soluble sugars, root density, and the number of root tips ranked from having the least to greatest effect on the classification of stumping and no-stumping.

**Conclusions**. The process of stumping may promote fine roots growth in *H. rhamnoides*, and is favorable for their longitudinal development. The fine root growing indices of *H. rhamnoides* responded positively to this process. Stumping promotes root activity and the creation of soluble sugar to maintain the growth and development of fine roots. It also inhibits the negative effects of proline and malondialdehyde on fine roots. Our study showed that the differential physiological indices were more important for classification than the differential growing indices.

## INTRODUCTION

Roots are among the key organs used by plants to exchange matter and energy with the external world (*Dannowski & Block, 2005*; *Bao et al., 2019*; *Deng, Guan & Zhang, 2018*), The morphology and physiological functions of fine roots can change to help a plant more effectively acquire resources (*Caldwell & Richards, 1986*). In the majority of forest ecosystems, the fine roots decreases with the increasing soil depth (*Song et al., 2020*; *Guo et al., 2021*; *Janna et al., 2021*). *Zewdie, Fetene & Olsson (2008)* studied the fine root distributions of two clone species of *Enset vebtricosum* in southern Ethiopia, and found that rainfall significantly affected the fine roots in the topsoil. *Vogt et al. (1996)* collected and analyzed a large amount of data and found that climate factors and nutrient conditions were important factors in deciding fine root biomass, while fine root production was mainly determined by the soil nutrient composition. Many researchers have analyzed the variations in fine roots by adjusting external factors (*e.g.*, soil nutrients, soil moisture, organics directional movement, and growing status of trees). However, there is little research showing the effects that changes to the plant have on the distribution and structure of fine roots.

*Hippophae rhamnoides* is a unique and dominant pioneer tree species found in feldspathic sandstone areas.It has a healthy root system with a strong sprouting ability and resistance against barren soils. However, owing to the special soil properties and geological conditions as well as drought and water shortages in feldspathic sandstone areas, the *H. rhamnoides* forests grown there have reduced growth and productivity after about 10 years (*Tian et al., 2021*; *Zhou, 2002*). Stumping, a common technical measure used in forest management and construction, may prevent ageing and decay and quicken the renewal and rejuvenation of plants. *Li et al. (2008)* found that stumped trees like *Calligonum mongolicum* have better growth and show significant biological benefits. *Dang (2012)* studied the stumping and rejuvenation of H. rhamnoides forests and found that stumped trees grew better than non-stumped trees, and that trees that sprouted after stumping did not suffer severe drought stress during the growing season. Stumping may promote the sprouting and renewal of plants by eliminating apical dominance and thus significantly decreasing the apical auxin concentrations so that auxin flows to the lateral buds and makes the meristems of the lateral buds grow (*Bowen & Pate, 1993*; *Pate et al., 1990*). The stumping height has been shown to affect the growth and production of roots (*Liu, 2018*). *Zhang et al. (2021)* found a stumping height of 10–20 cm significantly improved the diffuseness, absorption capability, and quantity of the roots of *H. rhamnoides* in China. *Liu et al. (2021)* compared the root fractal dimensions of *H. rhamnoides* at different stumping heights, and found that the stumping height of 15 cm was more favorable for root growth. The aboveground and underground part of a plants are an organized whole. After the aboveground portion is stumped, the fine roots undergo a series

of changes. With *Caragana microphylla* and *Salix gordejevii* Chang et Skv, stumping may increase the near-surface roots and biomass, especially for fine roots (*Cui & Liu, 2012*). Existing studies have focused on the matter transfer from roots to support the growth of the aboveground part after it has been cut or destroyed in sprouting plants (*Bowen & Pate, 1993*; *Van der Heyden & Stock, 1995*). There has been less research focused on fine root morphology and the effects of stumping on the growing or physiological indices of fine roots are unclear. Thus, further research on this issue will help us understand how the indices of fine roots in *H. rhamnoides* respond to stumping.

We hypothesize that the distribution of fine roots of *H. rhamnoides* after stumping in the soil will be similar to that of non-stumping; after stumping, the aboveground part of *H. rhamnoides* will quickly decrease, which will have a direct or indirect impact on the fine roots; and some indexes may change before and after stumping. In order to verify these hypotheses, we studied the distribution of fine roots and the changes of fine root characteristics after the implementation of stumping measures, and screened the difference indexes by PLS-DA to reveal the response of fine roots to stumping.

## MATERIALS & METHODS

### The study area

The study area of this experiment was located in the Geqiu groove watershed of the feldspathic sandstone areas in Ortos Plateau, China. The geographical coordinates of the basin are 39°42′N−39°50′N 110°25′E−110° 48′E and the average altitude reaches 800–1,590 m. The basin belongs to a typical moderate-temperature semi-arid continental monsoon climate zone with an average annual precipitation of about 400 mm, a frost-free period of 148 d, and an average annual temperature of 6.2–8.7 °C (*Wang et al., 2022*). The soils in this area are mainly loess, with a thin topsoil, loose soil structure, undulating terrain, crisscrossing gullies, and serious soil erosion. This watershed is mainly planted with artificial vegetation for soil and water conservation wind prevention and sand fixation. The major afforestation species include *H. rhamnoides*, *Pinus tabuliformis* Carr., *Caragana korshinskii* Kom., *Medicago sativa* Linn., and *Prunus sibirica* Lam.

### Methods

#### Experimental design

We selected 15-year-old artificial forests with a plant density of 2 m × 4 m in feldspathic sandstone areas for our experimental plots. The test field was located on the northwest slope at an approximately 4° slope. The sample sites were chosen with consideration for similarities in the forest ages, stand structures, and planting densities. In April 2020, we stumped the man-made forests 15 cm above the ground surface; a forest site without stumping was used as the control. The sampling sites were 150 m × 50 m in area. Each treatment was conducted in triplicate. Stumping was conducted using electric saws and pruning shears, which ensured that the incisions were flat and smooth without burrs. The complete stumping mode was adopted. To decrease moisture dissipation, the trees were painted after stumping. Then after 1 year of natural recovery, the basic conditions of the

**Table 1 Basic information for sampling plots.**

| Type | Parameter | Sample type | |
|---|---|---|---|
| | | C (Stumping) | CK (Non-stumping) |
| Stand factor | Average plant height (cm) | $87.12 \pm 1.96$ | $108.91 \pm 2.13$ |
| | Average crown width (cm) EW/SN | $63.32 \pm 1.58/61.01 \pm 2.04$ | $92.31 \pm 2.38/84.23 \pm 1.87$ |
| | Stem base diameter (cm) | $2.03 \pm 0.21$ | $1.29 \pm 0.17$ |
| | Length of branches grown in the year (cm) | $164.72 \pm 3.82$ | $83.15 \pm 2.28$ |
| | Forest land density(plant/m$^2$) | 1.12 | 1.18 |
| | Current survival rate/% | 82% | 89% |
| Chemical properties of soil | Total N/(g kg$^{-1}$) | $0.79 \pm 0.03$ | $0.57 \pm 0.02$ |
| | Total P/(g kg$^{-1}$) | $0.45 \pm 0.02$ | $0.22 \pm 0.01$ |
| | Organic matter/(g kg$^{-1}$) | $33.52 \pm 0.75$ | $24.73 \pm 0.80$ |

sample plots were surveyed, and the *H. rhamnoides* roots were sampled in the growing season (August 2021). The survey results are shown in Table 1.

Three plants were selected from each growing area with a height and canopy size close to the average values. Roots were sampled by combining the layered sampling method and soil auger method. Specifically, with the horizontal distance standard, the 0–50 cm soil layer was targeted in a 100 cm vertical circle. Three sampling clusters were selected, and the roots were collected using the 1/4 circle method (*Gao et al., 2020*). A 90° the fan-shaped area on the side with the lowest slope was chosen. After the ground litters was removed, the soils were divided into five layers by the depth of 10 cm each and horizontally into 10 layers of 20 cm each. Then samples were collected from each layer and were used to measure the growth properties of roots. The soil auger method was used collect the fine roots in the 0–50 cm layer at the adistance of 10 cm using the above sampling cluster as the center of circle. The five samples from the same standard cluster were mixed, and the fine roots were screened out and immediately stored in liquid nitrogen. These samples were used to measure the physiological properties of roots.

***Measurement of fine root growth characteristics***

All samples were taken back to the laboratory and placed in distilled water. Then the samples were loosened gently to discard soil particles. All fine roots with a diameter of less than two mm were picked out using vernier callipers and tweezers. The active roots were screened out according to shape, color, elasticity, appearance, and the centrum separability.

The selected fine roots were scanned under an EPSONV7000 root scanner, and the resulting images were analyzed using a winRHIZO root analyzer. Basic data including total root length (TRL), total root surface area (TRSA), total root volume (TRV), total root tips number (TRN) and total root projected area (TRPV) were obtained. After the scanning, the fine root samples were dried at 60 °C for 48 h, weighed, and the total fine-root dry weight (TRDW) was determined. The fine root characteristics were determined using the following parameters:

Root surface area density (RAD, cm$^2$ m$^{-3}$) = fine root area cm$^2$/soil volume m$^3$;

Root length density (RLD, m m$^{-3}$) = fine root length m/soil volume m$^3$;

Root volume density (RVD, $cm^3$ $m^{-3}$) =fine root volume $cm^3$/soil volume $m^3$;
Root number density (RND, $10^2$ $m^{-3}$) =root number/soil volume $m^3$.

### Measurement of fine root physiological characteristics

Proline concentrations were measured using ninhydrin colorimetry. Fine roots were weighed and added to a sulfosalicylic acid solution, followed by a water bath treatment. Then, the residues were filtered, and ice acetic acid and acidic ninhydrin were added, followed by a water bath treatment. Toluene was added to the mixed solution and the absorbance was measured after 5 min a spectrophotometer. The proline concentration was determined. Root activity was detected using chloridized triphenyl tetrazolium staining. A fresh root apex sample was weighed, and fully submerged in TTC and Tris-HCL for 2 h in the dark. An appropriate amount of sulfuric acid was then added to stop the reaction. Next, the root apex was removed, and the surface solution was dried. Ethyl acetate was added, the roots were ground, and the red liquids were (TTF) were extracted. After centrifugation, the absorbance was detected using spectrophotometry. The root activity was computed. Soluble sugar concentrations were measured using anthrone colorimetry. Fine roots were ground to a powder using ball milling and the powder, was screened and weighed. Then the roots were added into a distilled water bath and heated for 30 min. The supernatant was collected, centrifuged, cooled to room temperature, and diluted. Distilled water, anthrone ethyl acetate, and concentrated sulfuric acid were added to the tested solution. The absorbance was measured using a spectrophotometer after placement for five minutes. The soluble sugar concentration was determined and the malondialdehyde (MDA) concentration was monitored using the thiobarbituric acid (TBA) method. The fine roots were weighed, and TCA was added before the roots were ground. After the solution was centrifuged, the supernatant was extracted and added into a TBA solution. The mixture was heated in a water bath and centrifuged again. Then the supernatant was cooled to room temperature and the absorbance was measured using spectrophotometry. The MDA concentration was calculated.

## Data analysis

The differences in the growing indices and physiological indices between stumping and non-stumping and among different soil layers were tested *via* one-factor analysis of variance (ANOVA). The changes with statistical significance were further investigated *via* post-hoc pairwise analysis with the least significant difference method. The significance level was $P < 0.05$. The effects of the two variables (stumping, layer depth) on the growing indices and physiological indices were analyzed using general linear models and post-hoc analysis (the $S - N - K$ method). Statistical analysis was conducted on SPSS25. Plotting was performed on Origin 2021.

The stumping height classification (as determined by root growth and physiological composition) was evaluated using partial least squares discriminant analysis (PLS-DA). PLS-DA is a statistical method related to principal component analysis (PCA), and involves dimension reduction, regression modeling, and discriminant analysis. The model parameters included $y$-axis accumulative explanation rate ($R_{cum}^{2Y}$), model accumulative

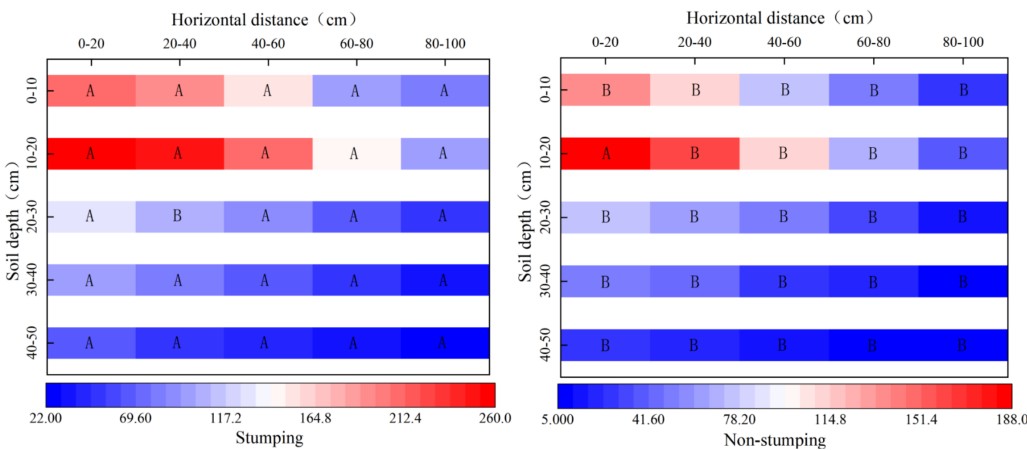

**Figure 1  Spatial distribution of the number of fine roots.** Note: different capital letters indicate significant differences among different stumping heights at the same soil layer ($P < 0.05$).

prediction rate ($Q^2_{cum}$), root mean square error of calibration (RMSEC), and root mean square error of cross validation (RMSECV). Each principal component provides a considerable amount of data for the variables. A difference between $R^{2Y}_{cum}$ and $Q^2_{cum}$ that is smaller and closer to 1, and RMSEC and RMSECV smaller than 1 and closer to 0 indicates that modeling effect is better (*Cheng et al., 2022*). The premise to use PLS-DA is that the prediction coefficient by PCA was close to 0.5. The data from different layers of each standard cluster were logarithmically processed to avoid errors due to too great a difference between the tested variables. Analysis was carried out on SIMCA and MetaboAnalyst 5.0.

## RESULTS

### Effects of stumping on spatial distribution of number of root tips

The number of root tips significantly differed between the two stumping treatments (Fig. 1). The horizontal distributions of the number of root tips were similar to those without stumping, as the number of root tips regularly decreased as the distance increased from the standard cluster. The horizontal fine root numbers with and without stumping ranged from 281–760 and 89–474, respectively, showing a significant difference. The proportion of the number of horizontal root tips at 60–80 cm and 80–100 cm within the standard clusters rose significantly by 1.96% and 4.32%, respectively, after stumping. In the vertical 0–50 cm soil layer, the number of root tips first increased and then decreased with the soil depth, regardless of stumping. The average number of root tips in each vertical soil layer varied from 211–957 after stumping, and 76–568 without stumping, revealing a significant difference. The number of root tips in the 30–40 cm and 40–50 cm layers rose significantly by 1.85% and 2.60% respectively, after stumping.

### Effects of stumping on fine root properties of *H. rhamnoides*

Stumping promoted the growth of the fine roots in *H. rhamnoides* (Fig. 2). Soil depth had similar effects on the root growing properties as determined by the soil depth for both

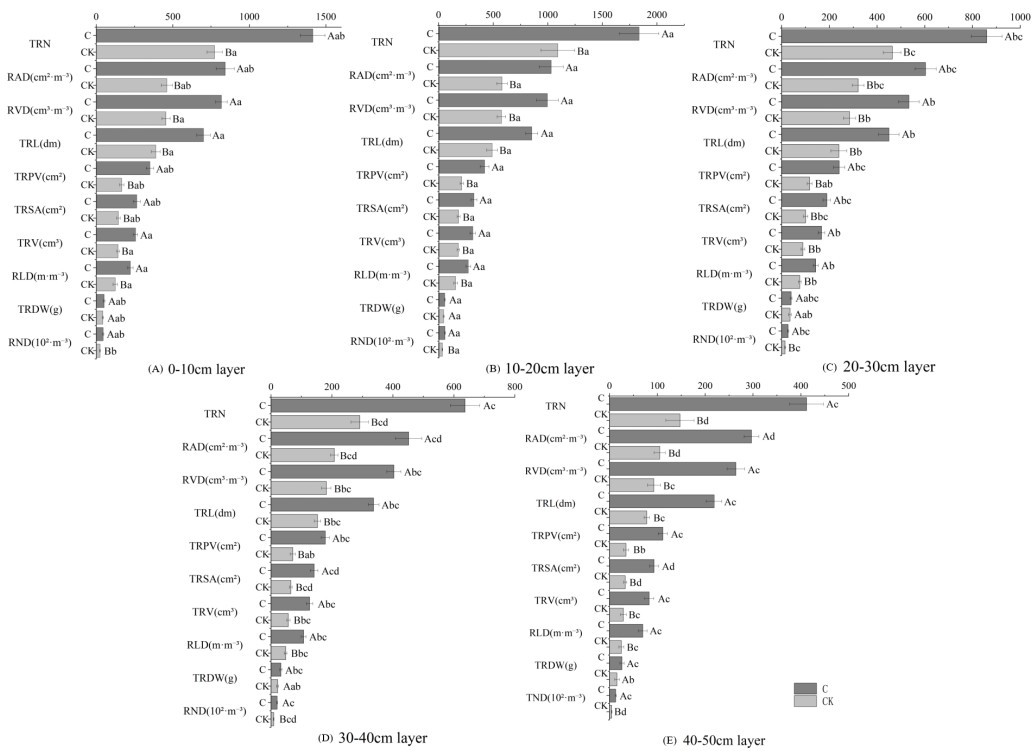

**Figure 2  Fine root growing properties of *H. rhamnoides* at different soil layers.** Note: different capital letters indicate significant differences among different stumping heights at the same soil layer; different lowercase letters indicate significant differences among different soil layers at the same stumping height ($P < 0.05$). TRDW, total fine-root dry weight; RND, root number density.

treatments. The fine roots were mainly distributed in the 0–30 cm layer and changed in the 'single peak' along with soil depth. The growing properties peaked in the 10–20 cm layer and then decreased with soil depth. Multivariate analysis based on general linear models showed the physiological indices of fine roots were significantly improved in the stumped groups compared with the non-stumped group. The growing indices of fine roots were most affected in the 10–20 cm layer and were least affected in the 40–50 cm layer. Moreover, the growing indices of fine roots responded to stumping to different degrees. In the 0–50 cm soil layer, the TRDW rose by 27.6% compared with the control, however, this was not significant. The other indices were significantly different from the control. The growing indices of the fine roots in different layers had varying degrees of response stumping. Compared with the control group, stumping increased the proportion of growing indices of fine roots in the 30–40 cm and 40–50 cm layers. These results indicate stumping may effectively improve the fine root properties of *H. rhamnoides* in deep soils and is favorable for the longitudinal development of fine roots.

## Effects of stumping on fine root physiological properties of *H. rhamnoides*

Stumping is more constructive to the physiological properties of fine roots and this practice promotes fine root development (Fig. 3). The proline, soluble sugar, and MDA concentrations in fine roots all first declined and then rose with the soil depth, regardless of treatments. These effects were minimized in the 10–20 cm layer. However, the changes in root vitality were shown to be opposite of this effect. Multivariate analysis based on general linear models showed that stumping significantly improved soluble sugar and RV, while no-stumping improved proline and MDA concentrations. The proline, soluble sugar and MDA in the 40–50 cm layer and the RV in the 10–20 cm layer were maximally affected. In the 0–50 cm layer, the physiological indices of fine roots responded to stumping to different degrees, and stumping significantly and negatively affected proline and MDA levels, which declined by 40.95% and 55.32%, respectively. Stumping also significantly and positively affected RV and soluble sugar, which rose by 68.58% and 36.87%, respectively.

## PLS-DA of stumping heights

The PCA results after different treatments are shown in Fig. 4A. C and CK overlap slightly, PC1 was 91.4%, and PC2 was 4.3%, with the total variance of 95.7%. PLS-DA analysis showed that C and CK are completely separated, and the discriminant components 1 and 2 accounted for 91.2% and 3.9% respectively. The results of PLS-DA were better than those for PCA (Figs. 4A, 4B). In order to further verify that PLS-DA was not overfitted, the variables of the classification $Y$ matrix were randomly arranged 200 times for permutation test. Figure 5 shows that the $y$-axis intercept of the fitted straight line $R^{2Y}_{cum}$ is 0.0995, which accords with the requirement that it shall be smaller than 0.3 and indicates that the model is reliable. The $y$-axis distance of the fitted straight line $Q^2_{cum}$ is $-0.347$, adheres to requirement that it shall be smaller than 0.5, suggesting that the model is not overfitted. The p of cross-validation ANOVA was smaller than 0.01, showing that the model is valid.

The VIP of PLS-DA was introduced to further test the contributions of different classes. Generally, a larger VIP means a higher contribution to model classification (*Gao, 2020*). Herein, the indicators were screened by setting the threshold at VIP > 1. An indicator with VIP > 1 is considered significant and can be used as the differential marking index. The VIPs of different indicators are shown in Fig. 6. The VIPs of four indices were all larger than 1 and their contributions ranked as MDA > SS > TND > TRN. These four indices considerably affected the classification between stumping and non-stumping, and there were significant changes between the two treatments. The VIPs of all other indices were smaller than 1 and there was little change between the two treatments, indicating that these indices very slightly affected the classification between stumping and non-stumping.

## DISCUSSION

The unique geological conditions and drought/water shortage in feldspathic sandstone areas resulted in severe decay in the 10-year-old *H. rhamnoides* lands. Thus, stumping was adopted to stimulate the sprouting and growth of the above ground and underground parts of *H. rhamnoides*. Stumping is the most effective measure to ensure the development of *H.*

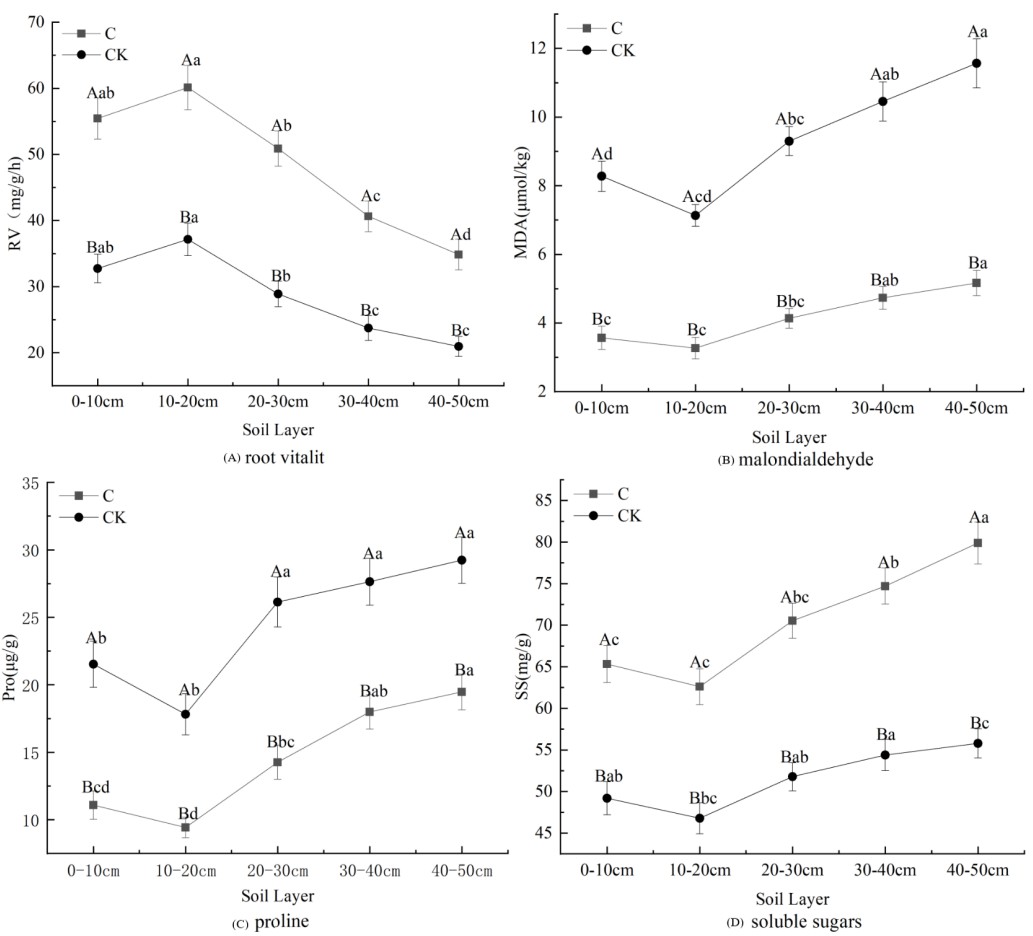

**Figure 3** **Fine root physiological properties of *H. rhamnoides* under different treatments.** Note: different capital letters indicate significant differences among different stumping heights at the same soil layer; different lowercase letters indicate significant differences among different soil layers at the same stumping height ($P < 0.05$).

*rhamnoide* for industrial purposes and to preserve water and soil in feldspathic sandstone areas. The various responses of fine root growth in *H. rhamnoides* stumping are complex processes. Roots allow plants to efficiently absorb water, nutrients and elements from soils. They are the major transfer routes for underground nutrients and fine roots are the greatest contributors toward this process (*Gao et al., 2020*). The growth and development of roots are jointly affected by the biological properties of plants and the environment. The number of root tips is a basic parameter of fine root research. Therefore, we measured the fine root spatial distributions of *H. rhamnoides* . Our results showed that the distributive laws for the number of root tips were similar between the two treatments, as the number of root tips regularly decreased with the prolonged distance from the sampling cluster. The number of root tips significantly rose after stumping in comparison with the control, and the proportion of the number of root tips at points farther away from the standard cluster increased(Fig. 1). Stumping was shown to promote the deeper growth of fine roots

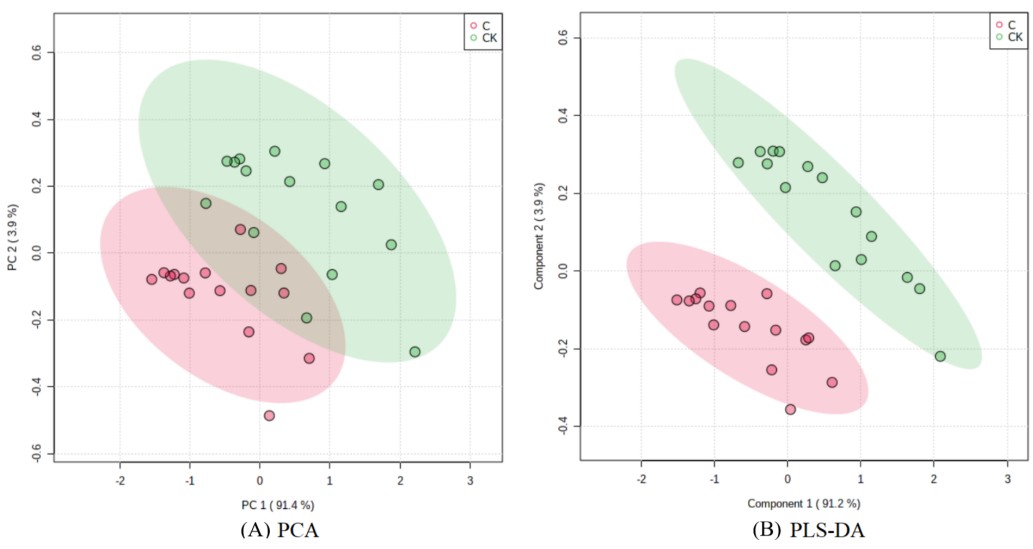

(A) PCA                    (B) PLS-DA

**Figure 4**   **Score maps of PCA and PLS-DA.**

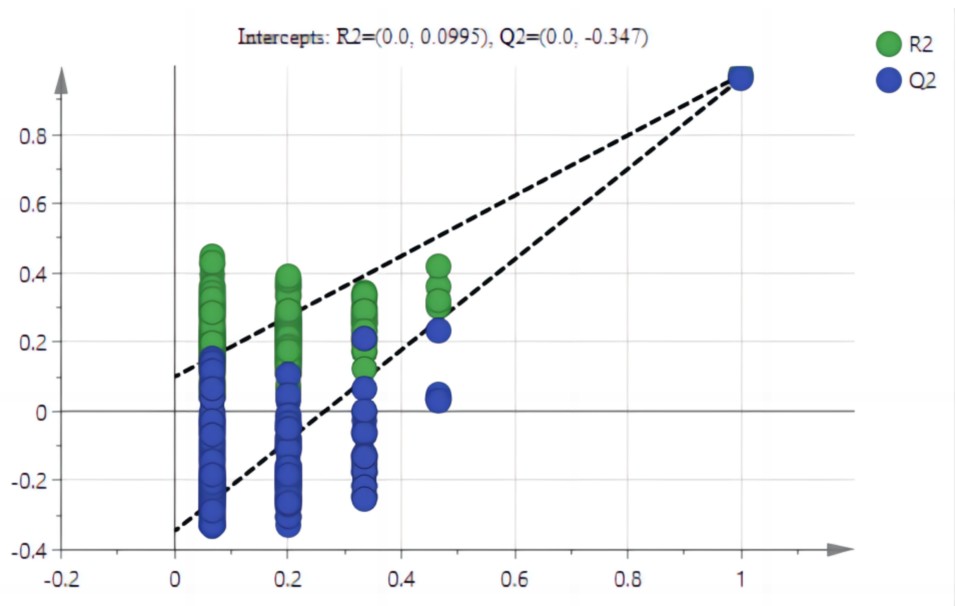

**Figure 5**   **Arrangement tests of PLS-AD mode.** Note: The $y$-axis accumulative explanation rate ($R^2$), model accumulative prediction rate ($Q^2$).

in both vertical and horizontal directions, which is consistent with the results of PLS-DA indicating that number of root tips and root number density changed significantly between the two treatments. In our study, the distributive laws of fine roots in the 0–50 cm vertical layer were similar between stumping and no-stumping. The RAD, RVD, RLD, TRN, RND, TRV, TRPV, TRSA, and TRTW of fine roots first increased and then decreased with soil depth, all samples were maximized in the 10–20 cm layer (Fig. 2) First, the fine roots

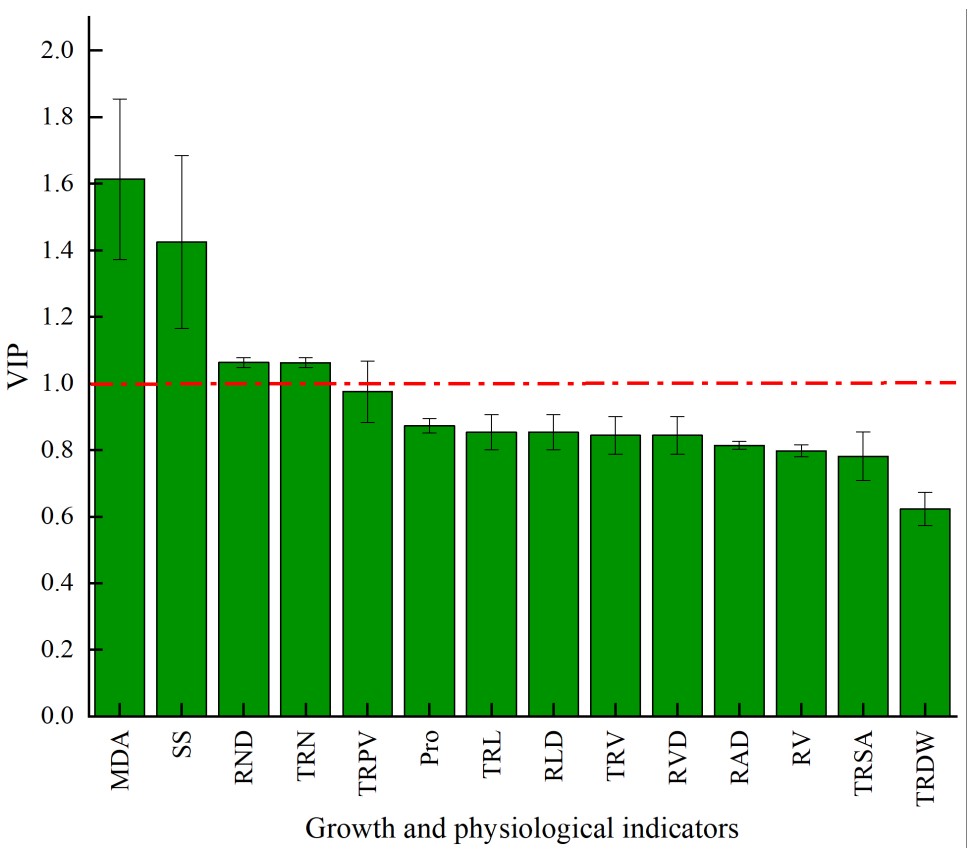

**Figure 6** VIP score chart for different indicators.

were concentrated in the 0–30 cm layer. The horizontal roots of *H. rhamnoides* sprout and propagate well (*Zhang et al., 2021*). Second, the topsoil has a smaller bulk density in relation to the deep soil (*Yuan, 2013*), and owing to the properties of feldspathic sandstone layers, the moisture in feldspathic sandstone areas mostly exists in shallow layers rather than deep layers (*Liu, 2021*). After one year of recovery from stumping, the fine roots propagated more successfully in stumped tress than unstumped trees, and the growth properties of the fine roots were significantly different between the two treatments, which was consistent with a previous study (*Zheng et al., 2010*). The proportions of fine roots in the 30-40 and 40–50 cm layers significantly increased after stumping in comparison with the control, indicating the stumping can promote the vertical growth of fine roots of *H. rhamnoides*, which is consistent with a previous study (*You et al., 2017*).

Root vitality reflects the oxidizing, reducing and synthesizing abilities of roots, and indirectly measures their metabolizing function (*Liu et al., 2016*). The fine root vitality of all Ping'ou hybrid hazelnut varieties was improved after stumping (*Shi et al., 2022*), which was consistent with our findings (Fig. 3A). Thus, root vitality was shown to be better after stumping and indirectly improved the growth of *H. rhamnoides*. MDA is produced from the oxidization of cell membranes, and its accumulation reflects the toxicity of reactive oxygen species. Thus, the variation in MDA concentrations revealed the degree

of damage sustained to the plant (*Cheng, 2012*). The present study shows that the content of MAD decreased afterstumping (Fig. 3B). It was previously reported that stumping effectively reduced the MDA concentration in the fine roots of *Tetraena mongolica* (*Wang, 2013*). Hence, stumping can effectively decrease the damaged degree of fine root cell membranes and reduce the damages to *H. rhamnoides*, thereby promoting its growth. The MDA concentrations in the roots of different drought-resistant maize seedlings increased with the intensification of water stress (*Qi et al., 2010*). The MDA concentrations in the seedling roots of *Glycyrrhiza uralensis* gradually rose with the intensification of drought stress, but decreased after stumping (*Liang & Shi, 2006*), suggesting that stumping can weaken the response to soil moisture stress in *H. rhamnoides* forests. Proline can stabilize enzymatic activity and protein concentrations in plants within certain ranges, and is an effective moderating substance *in vivo* (*Wang et al., 2007a*; *Wang et al., 2007b*). The proline concentration *in vivo* of plants is negatively correlated with soil moisture. Plants that are under drought stress synthesize proline to resist drought stress (*Shan et al., 2015*), which ensures their normal growth and development. When moisture stress reaches a certain degree, the proline concentration drops with the intensification of soil moisture stress. The proline concentration in fine roots is very low when no soil moisture stress occurs. The present study shows that the proline concentrations in fine roots of *H. rhamnoides* were lower after stumping (Fig. 3C). Together with the relationship between MDA concentration and soil moisture stress, it can be concluded that the decline of proline concentration is caused by the weakened soil moisture stress. Soluble sugars are considered as the basis for the revegetation of plants (*Wang, Yang & Wang, 2005*). *Zhu & Sun (1996)* found as the grazing intensity was increased, the soluble sugar concentrations in roots also gradually rose (*Wang et al., 2007a*; *Wang et al., 2007b*), which is consistent with our findings (Fig. 3D). This effect may be seen because the growth of the ground part is severely disturbed and to maintain growth and development the plants transfer nutrients to the roots to prevent nutrient loss. Thus, the soluble sugars that accumulate in the roots can guarantee the normal growth of the ground part so as to maintain the rapid recovery and growth of plants. Among all physiologic indicators, the MDA and soluble sugar contents of fine roots were shown to change the most significantly between the two treatments.

## CONCLUSIONS

We found that the fine root distribution first increased and then decreased both before and after stumping. Stumping can effectively improve the fine root properties of *H. rhamnoides* in deep soils and is favorable for the longitudinal development of fine roots.

Stumping was also found to promote the activity, soluble sugar concentrations, and all growing indices of fine roots, but inhibited proline and MDA concentrations, thereby alleviating the negative effects of proline and MDA on fine roots. Thus, stumping is more constructive to the growing and physiological properties of fine roots and promotes fine root development.

Stumping and non-stumping were defined as two distinct classes and the differential indices between the two classes were identified among MDA, soluble sugars, root density,

and number of root tips by using PLS-DA. The differential physiological indices are more important for classification than the differential growing indices.

### Funding

This study was funded by the National Natural Science Foundation of China (31500584), the Inner Mongolia Autonomous Region Applied Technology Research and Development Foundation Plan (2021GG0085 and 2021GG004), the Natural Science Foundation of Inner Mongolia Autonomous Region (2022MS03029), the Major Ordos City projects (2021EEDSCXQDFZ011), and the Inner Mongolia Ordos Application Research and Technology Development Project (2021YY SHE 106-55). The funders had no role in study design, data collection and analysis, decision to publish, or preparation of the manuscript.

### Grant Disclosures

The following grant information was disclosed by the authors:
National Natural Science Foundation of China: 31500584.
Inner Mongolia Autonomous Region Applied Technology Research and Development Foundation Plan: 2021GG0085, 2021GG004.
Natural Science Foundation of Inner Mongolia Autonomous Region: 2022MS03029.
Major Ordos City projects: 2021EEDSCXQDFZ011.
Inner Mongolia Ordos Application Research and Technology Development Project: 2021YY SHE 106-55.

### Competing Interests

The authors declare there are no competing interests.

### Author Contributions

- Haoyue Wang conceived and designed the experiments, performed the experiments, analyzed the data, prepared figures and/or tables, and approved the final draft.
- Wei Qi conceived and designed the experiments, analyzed the data, authored or reviewed drafts of the article, and approved the final draft.
- Yuefeng Guo conceived and designed the experiments, authored or reviewed drafts of the article, and approved the final draft.
- Yajie Xu performed the experiments, prepared figures and/or tables, and approved the final draft.

### Data Availability

The raw data is available in the Supplemental Files.

### Supplemental Information

Supplemental information for this article can be found online at http://dx.doi.org/10.7717/peerj.14978#supplemental-information.

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
