# Peer review of "Effects of stumping on fine root architecture, growth, and physiology of Hippophae rhamnoides"

_PeerJ, doi:10.7717/peerj.14978_

## Round 0.1 · original submission · Major Revisions

I have now the comments from three reviewers. All reviewers found the manuscript interesting, but they also raised some comments. Upon my own reading, I concur with these. Thus, I recommend that the authors undertake further revision in response to these comments.

Reviewer 1 ·

Basic reporting

The English language should be improved.
L44 MDA, SS, TND and TRN should be provided with full names.
L163, what unit root number is
I understand that this study has strong regional characteristics according to your references. From the MS, I know that stumping is a simple and practical technology for revegetation recovery. And I wonder if the related researches are conducted in other regions. The broader references will contribute to improve implications of your study.
L218 4. PLS-DA of stumping heights: this part should be greatly compressed and most of them are suggested to move to data analysis.
Figures are relevant to the content of the article, however, some legends should be appropriately described and labeled. For example, Fig. 1 and Fig. 5.
Moreover, some revisions should be made in Table 1. For example, Chinese text, the units of some properties.
I suggest to provide some pictures of stumped plots in supplemental files.
The MS did not give any relevant hypotheses.

Experimental design

I think the more details and sufficient information for plots and treatments should be provided. For example, the current woodland density, the number or area size you stumped, etc. When plants are stumped, whether you take some protective measures for cuts. And what survival rate of stumped plants.
The reason for “15 cm” stumping should be explained, what biological or physical mechanism?
Also, you had a treatment of stumping in this study, so “15 cm” can be deleted.

Validity of the findings

On the findings, I have a concern. You say that the lower proline is related with the lower soil moisture stress after stumping. while you also find the increased soluble sugars, which can improve the drought resistance. I can't quite understand their contrast responses. And what reasons for the increased soluble sugars are after stumping.

Additional comments

no comment

·

Basic reporting

This paper studied the effect of 15 cm stumping on root growth and physiological traits of Hippophae rhamnoides in Ordos. The author showed 15 cm stumping had a positive effect on root regeneration through investigating a range of quantitative and physiological characteristics of roots across multiple spatial gradients in vertical and horizontal directions. These findings are constructive to guide the vegetation restoration and reconstruction in desertification areas that suffer serious vegetation degradation and soil erosions, such as Ordos. Yet, I have several major concerns regarding the writing, the experimental methodology, the presentation of results as well as their interpretations. First, the introduction section paid too much effort on the background description, lacking a reasonable formulation of the scientific question to be solved. This may be, how stumping affects root growth and physiological activity by regulating root traits? or other more precise and in-depth questions. Second, the author focused on the effects of 15 cm stumping, is this height more convenient to operate, or has other advantages in comparison with treatments in other height? In addition, in the methods section, a more detailed and clear description on the experimental design is needed. Date analysis section, if right, the author may intend to compare the difference in root traits between different layers. Please check the description carefully and express it correctly. Third, in the result section, although well described in text, the figures are too vague. Finally, there lacking a mechanistic and insight explanation regarding the effects of stumping on root growth and physiological traits. For example, the author emphasized the roots of H. rhamnoides can sprout and propagate strongly in shallow soil layer (e.g., 30 cm) and ascribe this phenomenon to difference in soil conditions, such as soil moisture, and other physical properties. Since the author collected soils in different soil layer, is there any direct data evidence support this speculation? Or, is there evidence from previous studies underlie this phenomenon? In addition, the conclusion section is just repeating the results, with no condensing and improving of the key findings. This makes the study seem meaningless. This can be, for example, a mechanistic understanding of the finding in this study, such as stumping can enhance H. rhamnoides restoration by promoting root growth, physiological activity and reduce physiological stress. Overall, there remain some gaps to improve before this paper can be accepted and addressing the comments above described per section could contribute to improve the manuscript accordingly.

Experimental design

No comment

Validity of the findings

No comment

Additional comments

Title: growth physiology? Please check this expression.
Line 19, 25: What does “growth physiology” refers to? Please provide the correct statement.
Line 119-121: Specifically……. There are many grammatic issues throughout the manuscript, please check and revised it carefully.
Line 121: “Three standard clusters were selected”, “the 1/4 circle method”. I am not sure whether these terms are scientific.
Line 133: fine root specimens, is “root samples”?
Line 124: 5years, may be “5 layer”?
Line 258: “fine roots make the largest contribution”. It is not a complete sentence.
Line 219-226: The PLS-DA description can be placed in the data analysis section.

Reviewer 3 ·

Basic reporting

The language of this work should be seriously improved. The reference format in the main text is incorrect. Be aware that only family name needed.
Abstract
‘root number’ –do you mean number of root tips?
The second point of ‘The fine roots of ….in the 0-30 cm vertical soil layers’ might be better put at the beginning.
It might not clear for readers what proline and malondialdehyde imply and how they represent physiological properties of fine roots.
There are too many results listed in the results subsection in the Abstract. So, better to simplify them and only deliver very important information to the audiences. Much effect will be needed to make it succinct. Additionally, these results as presented are rarely discussed and interpreted in a broader scope.
Introduction
The field background started with a watershed in Inner Mongolia, which was such a narrow and local topic. Better to strengthen the importance of your study by provoking topic of general interests and widespread concern.
No hypothesis is presented at the end of Introdution.

Experimental design

The methods used to determine root morphological and physiological properties seem okay. The measurements of proline, root vitality, soluble sugar and MDA may need more detail to be reproducible by another investigator.

Validity of the findings

As commented above, the Results section should be simplified and more succinct. Main and interactive effects of stumping and soil depth on the parameters should be determined and shown, as currently only multiple comparisons or paired t-test were labelled.
Conclusions can not be just a summary of your findings. It should present the mechanistic understanding of your findings, answer your hypotheses, and illustrate the implication of your findings.

---

## Round 0.2 · Minor Revisions

The manuscript has been improved, however, there are still many minor issues that need to be addressed. Please carefully revise the manuscript according to the reviewers' comments.

Reviewer 1 ·

Basic reporting

This revised ms addressed most of my concerns. However, there are so many editing errors. The clear hypotheses have been not given.

Experimental design

no comment

Validity of the findings

no comment

·

Basic reporting

The author and their colleagues have made great efforts to improve this manuscript. However, there are still many minor issues need to be address. A few examples have been listed below for your reference. In addition, it is a great pity that I cannot get relevant information from the reply letter about how the author made revision, although main text has marked.

Experimental design

No comments.

Validity of the findings

No comments.

Additional comments

Line 27 What “fine root quantity” refers to? Number of root tips per area? Please clarify.
Line 32-34 The author found fine root quantities increased by 1.85%, 2.60%, 1.96, and 4.32 at different soil layer of the vertical and horizontal direction. I noted that these increase in fine root quantities did not seems significant. So, it seems So, it seems pointless to show this result. Please address.
Line 36 What does “growing indices” means? Perhaps it should be “growth indices”.
Line 44 The author stated that stumping could promote fine-root growth in deep layers. My question is how did you derived this conclusion? And why did stumping promote fine-root growth in deep layer rather than shallow layer?
Line 58 Perhaps you can cite some latest references.

---

## Round 0.3 · accepted · Accept

All concerns have been addressed in this revision. I have no other comments. Congratulations!

The Section Editor said:

> I recommend writing more detailed figure captions, allowing the reader to understand the figure without reading the complete paper. The first sentence of the conclusions is also a bit confusing: (".. first increased and then decreased both before and after stumping"). Please consider clarifying the findings.